# Development of a Geogenic Radon Hazard Index—Concept, History, Experiences

**DOI:** 10.3390/ijerph17114134

**Published:** 2020-06-10

**Authors:** Peter Bossew, Giorgia Cinelli, Giancarlo Ciotoli, Quentin G. Crowley, Marc De Cort, Javier Elío Medina, Valeria Gruber, Eric Petermann, Tore Tollefsen

**Affiliations:** 1German Federal Office for Radiation Protection (BfS), 10318 Berlin, Germany; epetermann@bfs.de; 2European Commission, Joint Research Centre (JRC), 21027 Ispra, Italy; marc.de-cort@ec.europa.eu (M.D.C.); tore.tollefsen@ec.europa.eu (T.T.); 3Institute of Environmental Geology and Geoengineering, National Research Council, Rome 00015, Italy; giancarlo.ciotoli@igag.cnr.it; 4School of Natural Sciences, Geology, Trinity College, D02 PN40 Dublin, Ireland; crowleyq@tcd.ie; 5Department of Planning, Aalborg University, 2450 Copenhagen, Denmark; javiereliomedina@gmail.com; 6Department for Radon and Radioecology, Austrian Agency for Health and Food Safety (AGES), 4020 Linz, Austria; valeria.gruber@ages.at

**Keywords:** geogenic radon hazard index, geogenic radon potential, European map of geogenic radon

## Abstract

Exposure to indoor radon at home and in workplaces constitutes a serious public health risk and is the second most prevalent cause of lung cancer after tobacco smoking. Indoor radon concentration is to a large extent controlled by so-called geogenic radon, which is radon generated in the ground. While indoor radon has been mapped in many parts of Europe, this is not the case for its geogenic control, which has been surveyed exhaustively in only a few countries or regions. Since geogenic radon is an important predictor of indoor radon, knowing the local potential of geogenic radon can assist radon mitigation policy in allocating resources and tuning regulations to focus on where it needs to be prioritized. The contribution of geogenic to indoor radon can be quantified in different ways: the geogenic radon potential (GRP) and the geogenic radon hazard index (GRHI). Both are constructed from geogenic quantities, with their differences tending to be, but not always, their type of geographical support and optimality as indoor radon predictors. An important feature of the GRHI is consistency across borders between regions with different data availability and Rn survey policies, which has so far impeded the creation of a European map of geogenic radon. The GRHI can be understood as a generalization or extension of the GRP. In this paper, the concepts of GRP and GRHI are discussed and a review of previous GRHI approaches is presented, including methods of GRHI estimation and some preliminary results. A methodology to create GRHI maps that cover most of Europe appears at hand and appropriate; however, further fine tuning and validation remains on the agenda.

## 1. Introduction

Indoor radon (Rn) is understood as an important health hazard (e.g., [1]). Therefore, it has been increasingly the subject of regulation aimed to reduce radon exposure. For Europe, the key document is the EURATOM Basic Safety Standards (BSS; [2]; similar to the IAEA-BSS [3]) and much literature deals with the many aspects of environmental radon, as well as a number of international research projects, such as RADPAR [4,5], SMART_RAD_EN [6], Rn in Big Buildings [7], and Life-Respire [8]. Some research was initiated to directly support the development and implementation of regulation, while other projects are focused on complementary activities such as to deepen the understanding of Rn behavior in the environment, to develop tools to quantify Rn, from measurement to displaying its distribution in the environment, and to assess its radiological significance. Among recent large-scale projects, the European Atlas of Natural Radiation (EANR; [9,10]) plays a key role, as well as the EURAMET MetroRADON project [11], which is devoted to improving the quality assurance chain from Rn measurement to aggregated products such as Rn maps, which serve as decision tools in Rn policy. Large parts of the work for this paper were carried out in the framework of the latter project.

Indoor radon concentration, which is the target quantity of regulatory concern, is to a high extent controlled by infiltration of radon generated in the ground, known as so-called geogenic radon. While mapping of indoor Rn concentration has been under way for years (shown e.g., in the EANR), this task has turned out more complicated for geogenic radon. So far, no European map of geogenic Rn exists. Geogenic Rn is usually quantified by the *geogenic Rn potential GRP*, a local quantity that characterizes the susceptibility of a location to geogenic radon (e.g., [12,13,14,15]). 

A further development is the geogenic *Rn hazard index GRHI*, which we understand as a generalized complement and extension to the GRP. The GRHI is more flexible and can deal with data reality which usual GRP definitions cannot handle. Its main application is thought to be large-scale mapping, i.e., on a European scale, in contrast to small-scale characterization e.g., of building sites or medium-scale national maps, of which their objective is supporting legislative and administrative implementation of the tasks posed by the European BSS. 

The purpose of this paper is to summarize the current (early 2020) state of conceptualization and definition of the GRHI. We present a brief review of the most promising techniques and attempts used to estimate and map the GRHI. Additionally, glimpses of GRHI maps developed using different techniques are displayed without going into technical detail in this paper.

## 2. Concepts

### 2.1. Geogenic and Anthropogenic Factors that Contribute to Indoor Radon

Indoor Rn concentration is controlled by both natural and anthropogenic factors. Natural factors, defined as geogenic factors, are related to radon generation and transport in the ground (e.g., [16,17,18,19,20,21]), whereas anthropogenic factors relate to construction characteristics of a building, including building materials and usage patterns (e.g., [12,22,23,24]). Meteorological factors may be considered in relation to both geogenic and anthropogenic systems, insofar as they can influence Rn transport in the ground, migration and accumulation of radon in the indoor environment, and construction style and building occupancy patterns (e.g., [25,26,27,28,29]).

Geogenic factors depend on geology, soil properties, and hydrology. These factors show a geographical trend and a spatial structure [20]. More generally, when a variable spreads in space and exhibits a certain spatial structure, it can be defined to be a regionalized variable (ReV) [30]. Geological, geochemical, and soil properties are subject to geographical trends. From a mathematical point of view, we can assume that environmental variables, i.e., geological, geochemical, and soil properties, are regionalized variables with two complementary aspects: A structural aspect that reflects the regional characteristic of the phenomenon, i.e., the trend;A random aspect that is the partly spatially structured, partly unstructured variability from one point to another at a local scale around the trend.

The former component reflects variability not captured by the trend and the latter reflects data uncertainty and variability within distance resolved by the estimation grid. The quantity of regulatory concern in radiation protection is the long-term mean indoor Rn concentration, which will be denoted as IRC in this paper. For practical reasons, long-term mean is mostly understood as the one estimated over the largest natural cycle (excluding possible cycles on a geological time-scale), namely the annual one (often though, the annual mean is estimated from shorter measurements, e.g., over three months). On this temporal scale, meteorological factors become climatic factors, which show geographical trends and can, therefore, be considered ReVs. Their temporal stationarity is a matter of debate: whether climatic change will have an impact on IRC is unknown. If this was the case, the IRC would not have a stable long-term mean value. However, we assume that this effect, if it exists at all, is very small and negligible for the near future; we are not aware of literature on the topic. 

Finally, the spatial statistical properties of anthropogenic factors are essentially unknown, although their existence can be plausibly assumed. For example, to some degree, climate (variable with geography, hence regionalized) determines construction of buildings and lifestyle. Also, local geology and landform can be assumed to influence construction style.

The most studied geogenic factors are Rn source (i.e., the geogenic radon source), related to geochemical properties of a geological unit, and Rn transport, quantified by the factors that govern the radon movement in the subsurface (i.e., soil permeability, faults and fractures, hydrogeology, and pedology). These factors are combined into a quantity called geogenic Rn potential (GRP), which conceptually, is designed to quantify the movement of geogenic Rn toward the shallow environment, of which its availability is to be exhaled from the ground and infiltrate buildings (e.g., [31]). It is noteworthy that to what extent available Rn leads to an actual IRC depends on anthropogenic factors. 

The GRP is considered as the most important regionalized predictor of IRC, that is, the predictor that shapes the geographical variability of the IRC. Therefore, models have been developed that attempt to predict IRC conditional to the GRP. Anthropogenic factors are statistically assumed as the noise terms, which in geostatistical language, is termed the nugget effect. The nugget effect is the short scale randomness or noise in the ReV that quantifies the variability between samples at a very close space in the experimental variograms. This assumption is probably not entirely correct, but spatial statistics of anthropogenic factors affecting the IRC are poorly understood at present. First attempts have, however, been made to include climate as a predicting factor, e.g., [32,33]. 

Several operational definitions have been proposed for quantification of the GRP. The most popular seems to be the so-called Neznal-GRP [14],
GRPNez = (SRC-SRC_0_)/(−log_10_ k − 10)(1)
with SRC denoting soil Rn concentration (kBq/m³), k, gas permeability (m²), and log_10_, the logarithm to base 10. SRC_0_, a small value, has been originally introduced for statistical reasons, but is set to zero by many authors, e.g., [34]. In [14], it was set to 1 kBq/m³. The numerical value of GRPNez depends on the sampling protocol, e.g., sampling depth and collection period (grab sampling or longer-term collection). As an example, for applications outside the Czech Republic [35,36], the German GRP map [34] is also based on the Neznal-GRP, but applies a sampling protocol [37] that is slightly different from the original Czech one.

### 2.2. History of the Geogenic Radon Hazard Index

A comparatively new concept is the geogenic Rn hazard index (GRHI), which was conceptualized around 2010. It was motivated by the lack of empirical GRP data in most of Europe as sufficient SRC and permeability data exist only in a few countries, namely in the Czech Republic ([35,36] where the concept originated), in Germany [34], where the GRP is used as an IRC predictor to estimate Rn priority areas (RPA), in Belgium [38,39], and in parts of Italy [40], Austria [41], and Spain [42], as well as in a few other countries where there was also no intention to generate country-wide coverage. Some countries chose SRC itself as a risk indicator, e.g., Estonia [43] and others; for some, see [44].

First attempts towards developing a European GRP map were started around 2008 [45], but it transpired that a realization of producing such a map is more complex than initially thought. The reason was—and still is—that in the foreseeable future, no consistent GRP dataset with European coverage is available.

The concept of the GRHI arose from the need to calculate a quantity from whatever geogenic quantities are regionally available. The challenge is to ensure consistency between the GRHI estimates in neighbouring regions if estimated from different predictors. That is, values of the GRHI must be equal between regions with the same objective geogenic controls, but with different data (e.g., in one region, uranium concentration in topsoil and soil granulometry are available, whereas in another region, SRC, soil type, and ambient dose rate). In other words, maps with different input variables must be “stitched together” seamlessly. Optimal prediction of IRC was not envisaged in that first stage of designing the GRHI [45]. Instead, IRC was understood as one of the possible candidates for covariates.

The first attempt to calculate a GRHI was reported by [46,47,48]. A set of “transfer formulas” to transform point data of SRC (e.g., permeability, uranium concentration in the ground, and ambient dose rate), which are more widely available than SRC and permeability, was reported in [48] into a GRHI. More recently, [49] suggested the attribution of a weight to the classified continuous or categorical input quantities (i.e., the covariates) that reflects its relevance in contributing to the envisaged index. The normalized (ranging from 0 to 1) weighted “mean class” will provide the GRHI, conceived as dimensionless quantity. The weights are the correlations of a covariate with the GRP, estimated in regions where the latter is available (Figure 1). The values of the input variables were associated to a 10 km × 10 km grid, according to the European Atlas of Natural Radiation [9,10], classified in several classes (four, called A to D, in the schematic of Figure 1), and then a weighted mean of the classes was computed. Weights should depend on correlation with a target quantity (e.g., GRP, where available; these regions would serve to “calibrate” the algorithm) and on the reliability of the cell value, quantified by the number (*n*) of original data aggregated into a cell.

A variant without resorting to classification of variables, i.e., leaving numerical variables as they are, has been shown in [50,51]. Covariates were transformed into their distribution functions (percentiles) and weights were defined by their correlation with IRC or GRP. 

The application of an explorative statistical technique as performed via a principal component analysis (PCA) on several covariates was developed by [31], thus using the first PC as GRHI. Recent attempts ([32,33,52,53]) utilized machine learning (ML) methods, which are considered particularly powerful for “high dimensional” multivariate settings and in particular, also for confirmative statistical techniques such as spatial regression (i.e., statistical approaches with many predictors).

A certain paradigmatic shift occurred during work on the EURAMET MetroRADON project, which started in 2016. The idea of “sewing” GRHI, estimated separately in various regions out of regionally available quantities, lost prominence against the idea to rely on databases which are available with European coverage. The advantage is that the consistency problem disappears; the drawback is that regional coverage of a quantity may be denser than the global (European) one. This is the case, most importantly, of the SRC and permeability, which are only available in a few countries, but are certainly very important GRHI predictors (see Section 2.4). Another issue of the newer GRHI conceptualization concerns the roles that the IRC may play and its relation to the GRP (Section 2.4).

### 2.3. Concept and Desired Properties of the GRHI

The GRHI can be conceptualized in different terms:a quantity which measures the contribution of geogenic factors to the potential risk that exposure to indoor Rn causes;a quantity which measures the availability of geogenic Rn at surface level;a measure of susceptibility of a location or of an area to increased indoor radon concentration for geogenic reasons;a measure of “Rn proneness” or “Rn priorityness” (in the logic of the BSS) of an area due to geogenic factors; i.e., a tool to decide whether an area is RPA.

Desired properties of the GRHI are:(I)consistency, across borders between regions, characterized by different databases used for the estimation; this implies independence of the actual database used,(II)exhaustiveness, which should reflect as much as possible the available geogenic information;(III)simplicity, which should be simple to calculate;(IV)predictor of the IRC, which should be a valid predictor of the geogenic contribution of indoor Rn concentration. This is motivated by its very concept.

These properties can be fulfilled only partly to different degrees by different concepts and are even partly contradictory. 

### 2.4. A Taxonomy of Approaches to Define a Geogenic hazard Index

Over the years, several attempts to define a GRHI have emerged. In Table 1, a tentative classification with some examples is proposed. We identify two conceptually different approaches, termed A and B (see Figure 2), and two variants, denoted by (1) and (2), referring to the exploitation of predictor quantities.

Approach A: Shortcut “geogenic”, attempts to construct the GRHI as combination of geogenic quantities such as geochemical concentration, lithology, and soil properties. Some variants include the IRC, motivated by the fact that the IRC also reflects, to some extent, geogenic radon. A combination is performed such that the resulting GRHI represents as much as possible the spatial variability of what is understood as quantifying the availability of geogenic radon for surface exhalation and infiltration into buildings. 

Approach B: “Optimal ~ IRC” combines the geogenic variables such that the combination best predicts indoor radon, meeting given criteria. The GRHI is the predicted value, optionally normalized e.g., to [0, 1]. Deviations between predicted and observed IRC are owed to data uncertainty (predictors and IRC), model uncertainty, and additional non-geogenic, i.e., anthropogenic controls of the IRC. The logic is summarized in Figure 2. In all cases, the models are built from all predicting data available in a domain. In some versions, only regions with sufficient data are used for model building. 

Variant (1): “Global” or “bottom up”, means that the model can be applied only at locations where all predictors and response variable are available. This is typically the case for regression models and models based on physical reasoning. Global models produce consistent results (property I, see Section 2.3) by default.

Variant (2): “Local” or “top down”, denotes models that can also be applied if regionally or locally, only some predictors are available. Consistency of results between regions in which different sets of predictors are available is the big challenge of this variant.

Approaches A (geogenic) have in common that a kind of weighted mean of predictors is constructed. The weighting may be implicit if in bi- or multi-variate scoring combinations of levels of categorical predictors are assigned certain GRHI levels. Often this seems to be done based on experience about the influence of a certain predictor. In other cases, the weights are defined as correlation coefficients between predictors, via principal component analysis (PCA), or by hierarchical analysis (SMCDA).

Approaches B can be characterized as generalized regressions; among them, traditional multivariate linear regression, general linear model (including categorical predictors), and machine learning (ML, among them, MARS, random forests, and support vector machines).

The desired properties, Section 2.3., are fulfilled to different degrees by these approaches and their variants:

The consistency property (I) is automatically fulfilled by variant (1) in the domain in which it is defined. For variant (2), this remains the crucial challenge.

Exhaustiveness property (II) is easier to fulfill for variants (2) than for (1), because for (2), local databases can also be exploited. Whether they are depends on the sophistication of the model.

Simplicity (III) is difficult to achieve for high-dimensional datasets and if spatial modelling is included. Easy for empirical classification and simple regression models. 

Predictor of the IRC (property IV) is fulfilled by default by approach B since the models are defined, by virtue of the regression paradigm, as yielding optimal predictors; how good they are differs between models. For models according to approaches A, this has to be checked afterwards.

This is summarized in Table 2. 

## 3. Methods

### 3.1. The Geogenic Radon Potential Compared to the Geogenic Radon Hazard Index 

The strict GRP concept consists of building a variable that reflects the Rn generation and transport processes based on their physical knowledge. This quantity is understood as location specific and scale-dependent or, in geostatistical terminology, having a point or block support, e.g., the 10 km × 10 km grid cells used in the European Atlas of Natural Radiation. 

The physically most straightforward definition may be
GRP = SRC × k(2)
which is the advective Rn flux normalized to the pressure gradient through an interface. It neglects diffusive transport, which is fair except for soil with very low permeability. 

The most commonly used definition, the Neznal-GRP [14], already has some traits of the GRHI (type B) because it is derived from matching a combination of SRC and permeability, aggregated into classes, to classes of the IRC by a kind of regression procedure. However, mapping the GRP requires datasets of soil Rn concentration SRC and permeability k, which are only available in few countries, see Section 2.1 and Section 2.2. 

While the GRP is derived from physics of Rn generation and transport, encompassing SRC (representing Rn source) and k (representing Rn transport), the GRHI is an extension which takes advantage of whatever geogenic quantity is available to quantify Rn availability at the surface and its potential to infiltrate into buildings (Section 2.3). Thus, GRP definitions may be considered as a sub-set of GRHI definitions.

### 3.2. Databases

To our knowledge, databases available on the European scale, covering almost the entire continent, include:
Geological maps:
-OneGeology [71] (Developed by EuroGeoSurveys’ European Geological Data Infrastructure within the framework of the GeoERA programme, 2018);-IGME 5000: 1:5 Million International Geological Map of Europe and Adjacent Areas [72,73];-Map of the World karst areas [74];-Global Active Fault database (GAF) [75].Soil properties: LUCAS database [76]; the database includes the following quantities (among others): topsoil fine fraction (as proxy of the soil permeability); available water content (AWC) (proxy of the soil porosity), chemical properties [77]. Another database of soil information is SoilGrid, containing global data estimated on a fine grid by machine learning [78].Geochemistry: GEMAS [79] and FOREGS [80], from which European uranium, thorium, and potassium maps have been created during the work on the European Atlas of Natural Radiation ([9,10] and references there).Aquifers (International Hydrogeological Map of Europe (IHME) 1:1,500,000) [81].Ambient dose rate: Across Europe, more than 5000 automatic stations continuously monitor ambient dose rate (ADR) as part of national radiological emergency warning systems. The data are stored and displayed by European Radiological Data Exchange Platform (EURDEP) [82,83] and the EANR. Normally, the ADR represents the natural background, of which their terrestrial component ([84]) is mainly due to natural radionuclides U, Th (more precisely their progeny), and K. Therefore, ADR is a proxy to geogenic radon (see below). A problem is that the data originate from technically different systems of which their harmonization is difficult.

Some examples of regionally available databases are:Ambient dose rate (ADR): e.g., Spain [85], the Czech Republic [86], Portugal [87], part of Germany [88,89];Saturated soil water content: Germany [90];Groundwater recharge coefficient: Ireland [91,92];Airborne gamma ray spectrometry: Ireland (Tellus project [93]).

Legends of geological maps are often simplified into lithological units which show similar geochemical characteristics and can be merged even though they are characterised by different stratigraphic positions (for example, Jurassic and Cretaceous limestone). The geochemical merging of lithologies is necessary to have sufficient IRC or SRC sample size per geological unit or for computational handling. In an example shown in [94], 178 units of the One Geology map were simplified into 28 units following a scheme proposed by [95].

Given that Rn availability at the surface is physically controlled by Rn source and Rn transport, the estimate of the Rn source term can be reasonably obtained by using geochemistry and geology, as geochemical surrogate. The estimation of Rn transport is, however, more problematic. Although no European database of soil permeability exists, there is hope that soil properties, hydrogeology, and tectonics may serve as proxies of permeability or in general, to emulate the Rn transport in the ground.

Predictors can be exhaustive in the sense that at every point of the domain (e.g., Europe), a predictor value is available. This is typically the case for categorical predictors such as geology, which is available as a map covering the entire domain. Others are available as finite sets of discrete point samples, typical sets of measured soil, or indoor Rn concentrations, geochemical concentrations, ADR, etc. These are sometimes made exhaustive by geostatistics (interpolation) before they can be used further. Other methods have this geostatistical trait intrinsically, typically some machine learning methods. 

Conceptually, one distinguishes between proxies (or surrogates) and physical predictors (Figure 3). The latter are ones that are in a causal relationship with the target variable, e.g., uranium concentration in the ground as a physical direct predictor of SRC. Proxies are ones that are statistically related to the target, but not directly linked by physical causality. An example is terrestrial component (TGDR) of ambient dose rate (ADR) as Z_1_ in the figure, which is statistically related to IRC (=Z_2_) because both share the same predictor, namely the uranium content in the ground (Z_0_). However, both ADR and IRC are also influenced by other variables, e.g., ^137^Cs fallout and Th concentration in soil (Z_0_”’ and Z_0_”) influencing dose rate and ground permeability (Z_0_′), the IRC; therefore, their correlation is weak.

### 3.3. Estimation Methods

Whichever definition of GRP or GRHI and whichever approach is chosen, the problem remains to estimate these quantities at a certain location or area. Since they cannot be measured directly, they have to be calculated from other quantities. The focus is on extracting information from several, or in some methods, many, regionalized databases. Putting it most generally, at each target point (or spatial target unit, such as pixels or whichever mapping support intended) of the mapped domain, one obtains the GRHI value by combining available data appropriately, where the criterion for appropriateness is different for approaches A and B. With most methods, spatial (or location) dependence of GRHI(x) is implicitly assured by one of its predictors. However, some methods additionally include location (coordinates) as *explicit* predictors. In the case of type B approaches (optimal predictors of IRC), the GRHI would be defined as the model outcome, with the understanding that the residuals IRC (observed)—IRC (modeled) represent anthropogenic factors and factors not accounted for by the geogenic predictors. 

#### 3.3.1. Concepts Type A

##### Multivariate Classification

Levels of categorical covariates are combined into levels of the categorical target variable. For example, geological units are levels of the predictor “geology”, in this case, unordered levels—such a variable is called nominal; permeability classes are levels of permeability, in this case, ordered levels—the variable is called ordinal. The target variable can be, for example, GRP classes (ordinal). To a large extent, combination rules are empirical, based on experience. 

As an example, in the Czech Republic, a rule has been established to assess the risk class of a location based on cross-tabulation of classes of SRC and permeability [14]. The U.S. EPA [69,70] proposed a scheme incorporating IRC, geological evidence, permeability, U concentration (by airborne gamma ray spectrometry), and “architecture type” (kind of foundation). Missing data are possible, leading to lower confidence of the index value; therefore, the method has been classified into “local” in Table 1, where other examples are also quoted. 

##### Principal Component Analysis (PCA)

In a high-dimensional setting, such as for the prediction of geogenic Rn from many potentially predicting quantities, one would first attempt to identify the amount of information that the set of covariates actually contains; many of the predictors tend to be correlated between themselves, hence carrying redundancy. Principal Component Analysis (PCA) is a well-known, explorative method of which its main objective is to reduce the data complexity with minimal loss of information and to create a set of new uncorrelated variables (factors) linearly linked to the original ones. They are arranged such that most information is contained in the first or the first two or three factors. 

PCA has advantages and disadvantages. The advantages are: (i) there is no response variable and all variables are, in theory, of equal importance; (ii) it reduces the number of variables to be further considered.

The disadvantages are: (i) principal components as new variables are less easy to interpret than the original ones; (ii) there is no test to verify the goodness of the results; PCA is an exploratory analysis with subjective interpretation, although there are rules for reading the variables in the factorial space; (iii) the number of retained factors must be selected with great care in order to not discard essential (for a given objective) information contained in the original variables; (iv) in classical PCA, only numerical covariates can be included, but not categorical—in particular, nominal ones. Detailed descriptions of PCA technique can be found in [96] and references therein and [97].

The GRHI can be defined as the first PC or as a combination of a few components with highest weight. Regionalization is performed along the line explained at the end of the following sub-section. 

##### Transfer Models

A set of formulas or rules is established that transforms available variables into a GRHI; they are of the type GRHI = f (Y_1_, …, Y_n_); if predictor Y_i_ is not available, estimate it from different variables U_i_ as Y_i_ = f_(i)_ (U_1_, …, U_k_) and so on. Rules are look-up tables, which associate a level of a categorical variable with a needed Y_i_; (e.g., factor = geology, level i of this factor = L_i_ = quaternary sediment, which has Y_j_ = mean soil Rn concentration value y_j_ = 20 kBq/m³). The transfer formulas are deduced from studies about relationships between geogenic variables.

The idea is to take advantage of whatever data are available in a region. The evident problem is consistency between two neighboring regions, which are physically identical (same geology, same soil type, same geochemistry etc.), but in which different predictors are available and in which the GRHI therefore has to be estimated differently. The consistency problem is visualized in Figure 4.

Two ways of regionalization are conceivable, i.e., establishing the GRHI as spatial function GRHI(x) for every point (or spatial unit) x of the domain. (1) For discrete sample type predictors, estimate them at every needed point of the domain, usually by geostatistical means, and build GRHI(x) = f (Y*_1_ (x), …, Y*_n_ (x)), Y*(x) the interpolated value. Alternatively, (2), calculate GRHI(x_i_) at points x_i_, where predictors are available, and afterwards subject GRHI to geostatistics to obtain interpolated GRHI* (x) for every x.

##### Spatial Multi-Criteria Decision Analysis (SMCDA)

GIS-based (or Spatial) MCDA (SMCDA) is a set of procedures that can be used to combine criteria maps (i.e., variable layers) with respect to their relative importance and derive relative weights for the criteria [98,99]. In the context of this work, SMCDA involves combining and handling of different criteria that determine the presence of RPAs, then uses the Analytical Hierarchy Process (AHP) [100] to assess their relative importance and derives the weights for each criterion; then, the final suitability scores ([65]) are calculated by using the weighted linear combination (WLC) of original criteria maps [101]. A more elaborate example is shown in [65].

SMCDA is an explorative technique; it does not make use of a response variable and of validation techniques. SMCDA involves subjectivity (e.g., in choosing the criteria and defining the relative importance of each factor). Result validation can be provided by direct measurements and by sensitivity analysis ([102,103]). Some SMCDA versions can be understood as mathematically optimized multivariate classifications. The technique was developed to help decision makers in sustainability planning and to provide outputs to be easily understood by non-experts. The method has also been applied for finding best consensual solutions in cases of stakeholder conflicts, e.g., [104]. It could be that it can be applied to RPA delineation, including under the constraints of conflicting stakeholder interests, which is a big political issue as current experience shows.

For further resources, see e.g., [105,106,107,108,109] and the Wikipedia entry “Multiple-criteria decision analysis”.

#### 3.3.2. Concepts Type B

##### Multivariate Regression (MR)

Regression means, to find the expected value EZ of a response, dependent, or target variable Z, given (or: conditional to) one or several predictors or independent variables Y. This is done by minimizing a loss function, originally the sum of squared deviations of observations z from predicted or estimated E(Z | Y = y). The theory has been developed for two centuries, with abundant literature available, and shall therefore not be repeated here. Variants include categorical predictors (general linear model) and non-linear link functions between Z and Y and non-Gaussian error models (generalized linear regression); most importantly, logistic regression, aimed to predict a binary variable (a condition fulfilled or not) or a probability. Among important problems are collinear and nested predictors (i.e., the independent variables are dependent among them), which can invalidate analyses. Including location (coordinates) as predictor leads to the reasoning of geostatistics. Regionalization to obtain Z(x) for every point x in the domain proceeds along the lines described above. 

##### Machine Learning (ML)

This class of methods took their name from the idea that the physical structure that underlies a dataset (which can be understood as realizations of a true physical process) shall be recovered from the data themselves, without stipulating a model. The rationale is that in complex situations (many predictors or covariates, related among them, etc.), this model is not only badly known, but is actually difficult to write down explicitly because of its complexity. Instead, the algorithm identifies patterns in the data which are observable representations of the physical reality, which approximate the physical model by numerical decision rules. Once recognized, the patterns can be used to predict a response (e.g., IRC) from observed predictors (e.g., geology, uranium concentration in the ground, climate, …). In this sense, ML is a type of regression without a specified regression model. The conceptual difference against regression is visualized in Figure 5. A standard textbook on ML is [110].

In radon science, ML has first been used, to our knowledge, by [53,67] and [52] for spatial settings and by [111] in time series analysis. Current work at the BfS aims to improve regional GRP and IRC prediction by including high numbers (up to 100) of potential predictors [32].

ML offers the possibility to include location (parameterized by the coordinates) as covariates. For Rn estimation, trials at the BfS seem to show that this does not lead to improvement, probably because sufficient spatial information is already contained in the regionalized predictors.

## 4. Exemplifying Preliminary Results 

So far, no authoritative GRHI map exists on a European level. However, sevd m neral attempts have been made to explore the potential of different approaches. Some are shown in this section. Note that these are trials only, of which their objective is to acquire experience with methodology without authoritative relevance.

The maps shown in the following section have certain patterns in common, but also important differences. There may be several reasons for this, from lack of data to misspecification of the model structure or the algorithm.

Maps reported in Section 4.1 (Figure 6) and Section 4.4 (Figure 10) belong to approach A, whereas maps of Section 4.2 (Figure 8) and Section 4.3 (Figure 9) to approach B. 

### 4.1. Geological Classification

The very first trial was made by [112,113]. Geological units taken from OneGeology were coded or “calibrated” according the Neznal-GRP for units where data were available, mainly in the Czech Republic, Germany, and Belgium. Regions that could not be coded in this way have been left blank in Figure 6. Classes were defined deliberately.

Evidently, this approach suffers from (1) lack of data and (2) the fact that “extrapolating” from units where GRP information is available to nominally the same or geologically similar geological units, but without data, is questionable.

The general geographical pattern is very similar to the one of the European Indoor Radon Map [9,10], as of course it must be, but no correlation analysis or validation was attempted because this trial was a technical feasibility study only. Class 1 is the lowest and 4 is the highest GRHI. 

### 4.2. Multiple Regression

The result was first shown in [68]. Starting with about 100 predictors (database references in Section 3.2):Geochemistry: A combination of FOREGS and GEMAS databases, 59 elements; missing uranium values estimated by lanthanum and cerium because these elements are highly correlated; about 5000 data points in Europe.Soil properties: from LUCAS; point data projected to geochemical data points by geostatistics. Fine fraction tentatively defined as
FF = (clay + silt + 0.05 sand)/(100 + coarse fraction)(3)
as permeability proxy (the definition is debatable);Geology: IGME 5000.


Through trying (among others, by inspecting correlations between variables), for further analysis, the set of covariates was reduced to pH, TOC, FF, CF (coarse fraction), soil bulk density, ln(U), K_2_O, Al_2_O_3_, SiO_2_, Fe_2_O_3_, CaO, and geo1; with geo1 = {carbonate, meta-sediments, siliciclastics, Cenozoic sediments, basic igneous rocks, intermediate igneous, pre-Variscan acid igneous; Variscan acid igneous, post-Variscan acid igneous}. 

The target variable is AML of the European Atlas of Natural Radiation (AML = AM_cell_[ln(IRC)] = arithmetic mean of the logarithms of IRC within 10 km × 10 km cell), interpolated to geochemical locations, i.e., AML in hypothetical cells around these locations.

Applying a general linear model with stepwise elimination of irrelevant covariates (F-test) led to {geo1, FF, pH, bulk density, K_2_O, ln(U)} as the best predictor, which explains r² = 26% of variance. Inclusion of annual mean temperature would increase this to 29%.

The model f(Y)(x) (Y—vector of covariates, x – location) was subjected to ordinary kriging to the original Atlas cell locations and the results quantile rescaled to [0,1] by z ⟶ F_Z_(z). Different rescaling is equally possibly, e.g., by linear rescaling, z ⟶ (z–z_min_)/(z_max_–z_min_), tgh, or nscore transforms. The result is shown in Figure 7. In the map, 0 is the lowest and 1 is the highest GRHI.

### 4.3. Machine Learning

The algorithm Multivariate Adaptive Regression Splines (MARS) (an introduction can be found in [110] and [114]) creates piecewise linear models where each predictor models an isolated part of the original data. For this purpose, each data point for each predictor is evaluated as a split candidate by creating linear regression models. The contribution of the individual terms in the model is evaluated based on the generalized cross-validation (GCV) statistic. In this study, the implementation in the “earth” package [115] in R was used.

The target variable was AML (like above), but only 10 km × 10 km cells with *n* > 30 original indoor Rn data were used for training the model. The model was fitted using >100 candidate predictors using the model inherent predictor selection. The hyperparameters of the final model are degree = 1 (i.e., no interaction between variables) and nprune = 83 (i.e., 83 terms in the final model). The selected predictors comprise:Geology: IGME 5000: lithological unit (attribute “Portr_Petr”, 92 classes);Hydrogeology: IHME 1500 ([116]): attribute “Litho level 2” (85 classes);Soil: regions of Europe (285 classes) ([117]);Soil physical properties [76]: Silt content, Clay content, available water capacity, bulk density, coarse fragments;Soil hydraulic properties: hydraulic conductivity [118]: Saturated hydraulic conductivity (at depths 0 cm, 60 cm, and 200 cm);Location: Longitude and latitude.

The result is shown in Figure 8 (first in [68]). The calculated values were linearly rescaled to [0,1], like above. For multiple regression and ML, the pattern is very similar to the one of IRC, which was of course to be expected because IRC is the independent variable in the models. The ML method performed very well with r² = 0.52 between predicted and observed AM (IRC per 10 km × 10 km cell) of the test dataset (which has not been used for model building). Again, 0 is the lowest and 1 is the highest GRHI.

However, the model building procedure applied for ML in this study has some limitations, namely
(1)categorical predictor data (geology, hydrogeology, soil regions) could be re-classified with respect to Rn to reduce the classes and the risk of over-fitting.(2)no external predictor selection procedure was applied, only the model inherent predictor selection. This might result in the appearance of non-informative predictors in the final model and might cause over-fitting.(3)The cross-validation procedure in this study (stratified sampling) did not account for spatial auto-correlation in the data. This might produce a too optimistic r² as a consequence of spatial auto-correlation because test data might be within the correlation length of training data (see [119] for details). Therefore, independence between training and test data is not guaranteed. In newer versions (currently in work), spatial cross-validation is being implemented.

Further, it should be noted that other ML algorithms, especially ensemble techniques (e.g., random forest) might be more powerful than MARS for modelling a noisy target variable such as IRC. Nonetheless, the ML result presented in this study indicates the potential of ML for GRHI mapping and will be even more robust when the previously mentioned methodological specifications will be implemented.

### 4.4. Principal Component Analysis

Reference [31] explored dimensional reduction by PCA of the following set of variables:Geochemistry: GEMAS + FOREGS, U, Th, and K, as in the European Atlas of Natural Radiation.Soil properties: Fine fraction FF in topsoil from LUCAS, as in the Natural Atlas.Tectonic fault lines: global fault layer from ArcAtlas, ESRI; areal density.Earthquake epicenters: [120].Geothermal and volcanic areas: in terms of heat flow (the heat flow map of Europe has been obtained by analyzing the Global Heat Flow (International Heat Flow Commission of the International Association of Seismology and Physics of the Earth’s Interior, IASPEI).

Note that indoor Rn (IRC) is not among the variables, nor is soil Rn (SRC). All data were projected into the 10 km × 10 km grid of the European Atlas of Natural Radiation; map of the heat flow was obtained by kriging point data; maps of the fault and earthquake density were obtained by kernel density estimation; maps of the FF, uranium, thorium, and potassium were available from the database of the European Atlas of Natural Radiation. Values of the variables were assigned to the 10 km × 10 km grid centroids in order to obtain the dataset for the PCA. The raw (unrotated) PCA result is shown in Figure 9. One can recognize two essential groups: (U, K), which represent the source term and FF, faults etc., which represent transport properties. 

The GRHI at point x is defined as
GRHI(x) = ∑_(over variables j)_ w^(1)^_j_ y_j_(x)(4)
where y_j_(x)—value of variable j (e.g., U concentration etc.) at location x, w^(1)^_j_—loading of variable j in the first principal component = abscissa (F1) value in Figure 10.

The resulting GRHI is mapped in Figure 10. While the expected geographical pattern is partly apparent, it does not seem appropriate in other parts of Europe, notably Scandinavia, the Bohemian Massif, and the Pannonian Basin, if compared to the maps in Figure 7 and Figure 8. The difference is owed to the fact that it is generated by a different approach, namely A instead of B. 

## 5. Conclusions

Mapping geogenic radon appears to be neither a straightforward nor a technically easy task. The reasons lie in its definition; in particular, would we like to first capture the geogenic variability (approach A) or optimal predictability of indoor Rn (approach B)? Furthermore, concerning the estimation technique, which technique to use? How will various predictors be included? 

Different trials for approximately the last 10 years led to variably satisfying results, but in any case, served to gain experience with different approaches and techniques. The first version of the European Atlas of Natural Radiation did not include a European map of geogenic Rn because it was felt that the concept and techniques were not yet sufficiently developed. It seems that we are now converging towards a robust European geogenic Rn map, or perhaps several, reflecting different properties, represented by approaches A and B, which both have their justifications.

At the moment, it seems that of all the methods investigated, for approach A (“geogenic”), the most promising method is PCA, while for B (“optimal to IRC”), machine learning is most powerful, but methodologically has not yet been fully explored. However, further multivariate methods should be explored, notably spatial multi-criteria decision analysis for A and B and varieties of PCA, for approach A.

We hope that this work serves as an incentive for further research. We see two open fields:

Conceptual: Refinement of GRHI definitions; specify which definition serves which purpose. Probably different definitions will lead to different maps. In the end, different definitions should be given different names to avoid confusion.

Technically: improvements are certainly possible in existing methodology, but it would also not be a big surprise to see new methods appearing, given the current dynamic in radon science.

The main motivations behind conceiving the GRHI are (1) to create a unified measure of the natural availability of geogenic radon which can be estimated from different types of geogenic quantities and (2) to generate a methodically homogeneous European-scale map of geogenic radon. Consequently, methods and results shown here were tailored to exploit databases that cover most of Europe. However, there is no reason why the same rationale should not be applicable on a regional scale, possibly in higher resolution.

## Figures and Tables

**Figure 1 ijerph-17-04134-f001:**
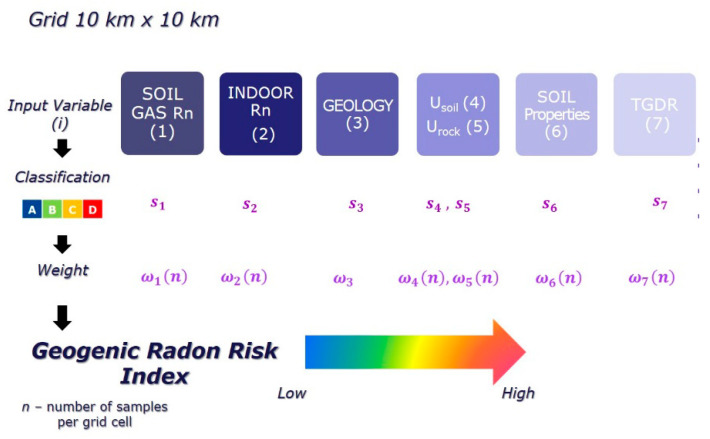
General workflow of multivariate classification approach to construct a geogenic radon hazard index (GRHI) [49]. TGDR—terrestrial gamma dose rate.

**Figure 2 ijerph-17-04134-f002:**
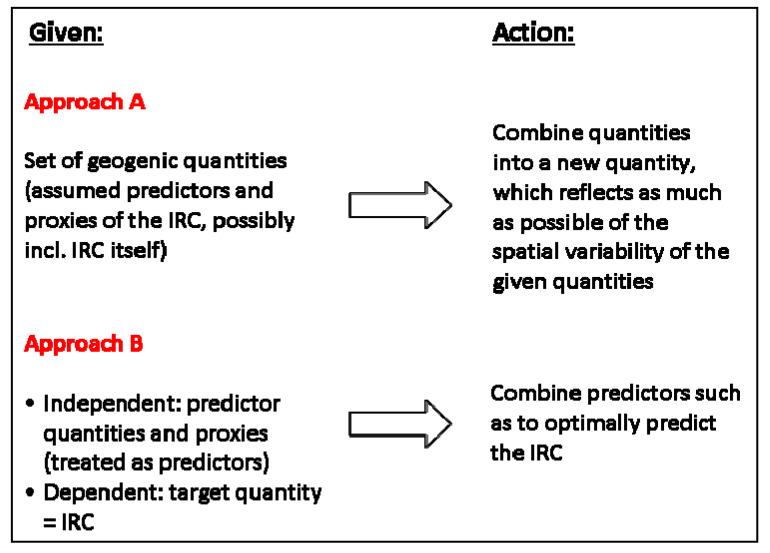
Approaches A and B.

**Figure 3 ijerph-17-04134-f003:**
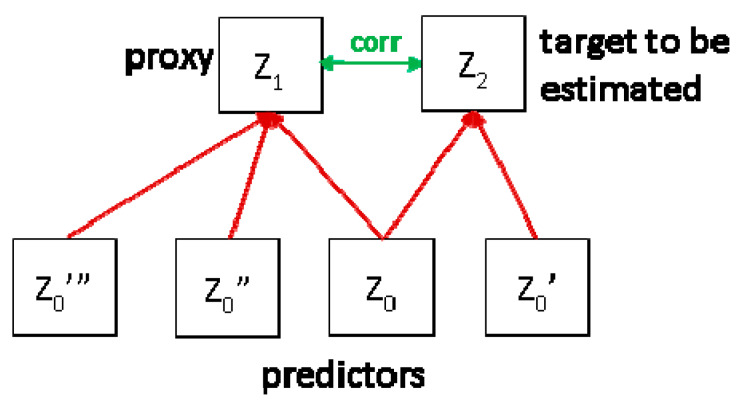
Physical predictors and proxies (see text).

**Figure 4 ijerph-17-04134-f004:**
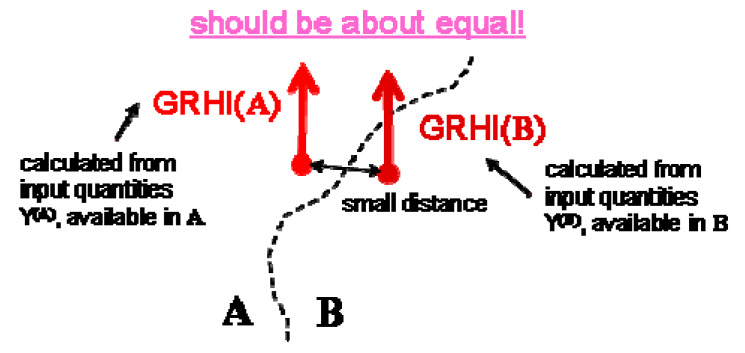
Consistency between quantity GRHI calculated in regions A and B from different sets of predictors, Y^(A)^ and Y^(B)^.

**Figure 5 ijerph-17-04134-f005:**
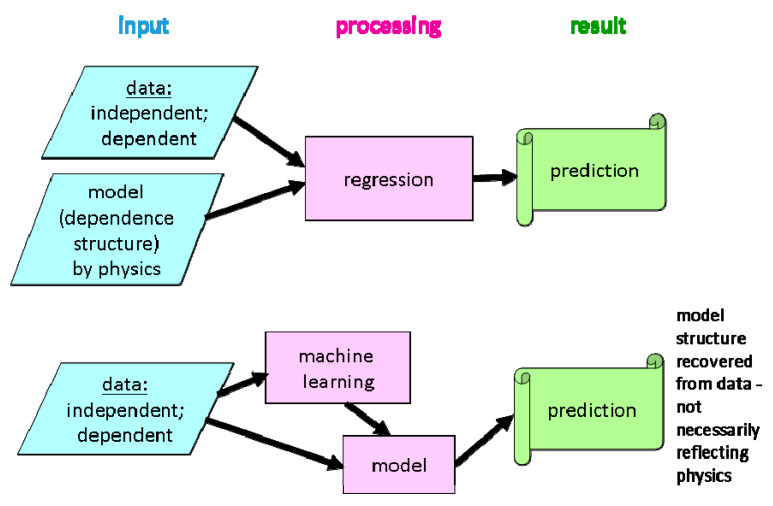
Conceptual difference between classical (generalized) regression and machine learning.

**Figure 6 ijerph-17-04134-f006:**
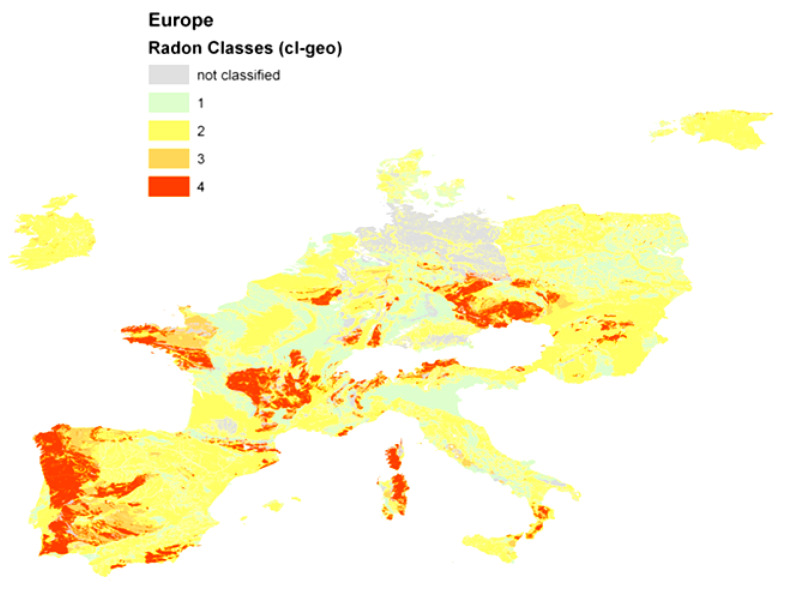
Classification of geological units according to the Neznal-GRP; from [112].

**Figure 7 ijerph-17-04134-f007:**
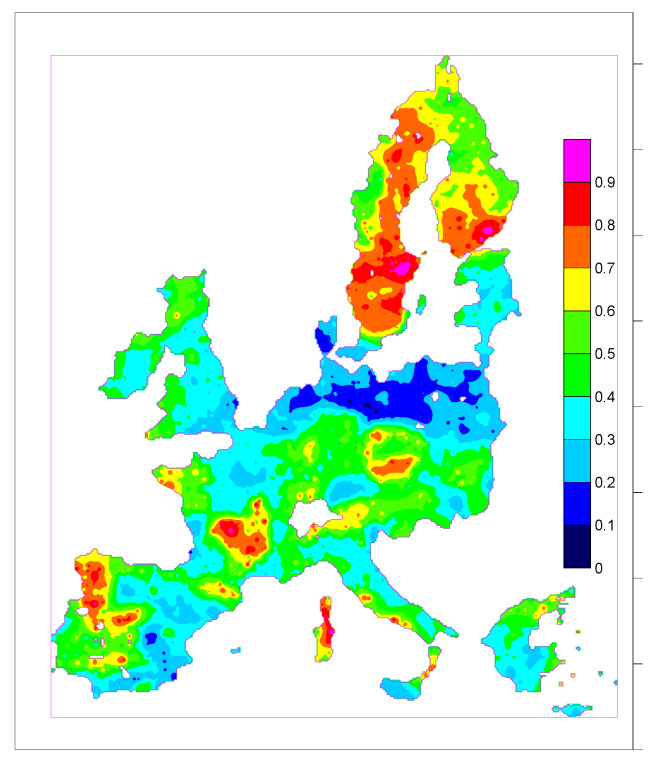
GRHI map created by multiple regression (from [68]).

**Figure 8 ijerph-17-04134-f008:**
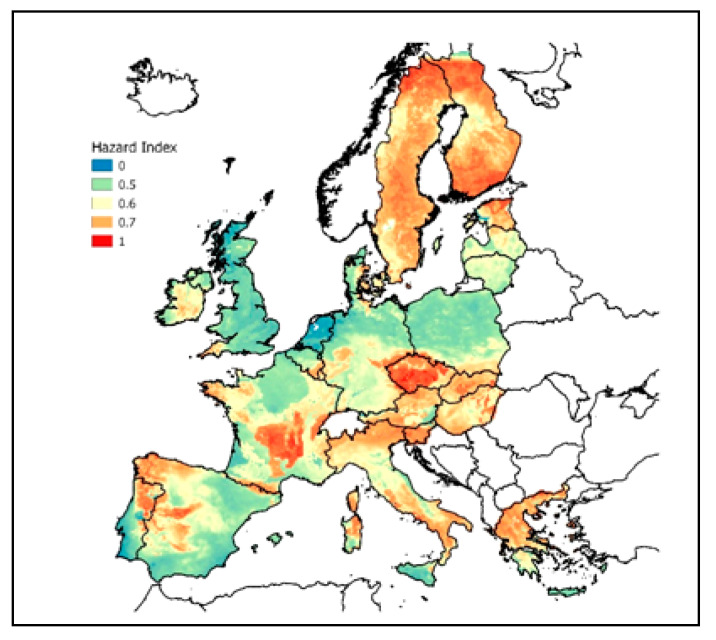
GRHI map created by machine learning (MARS) (from [68]).

**Figure 9 ijerph-17-04134-f009:**
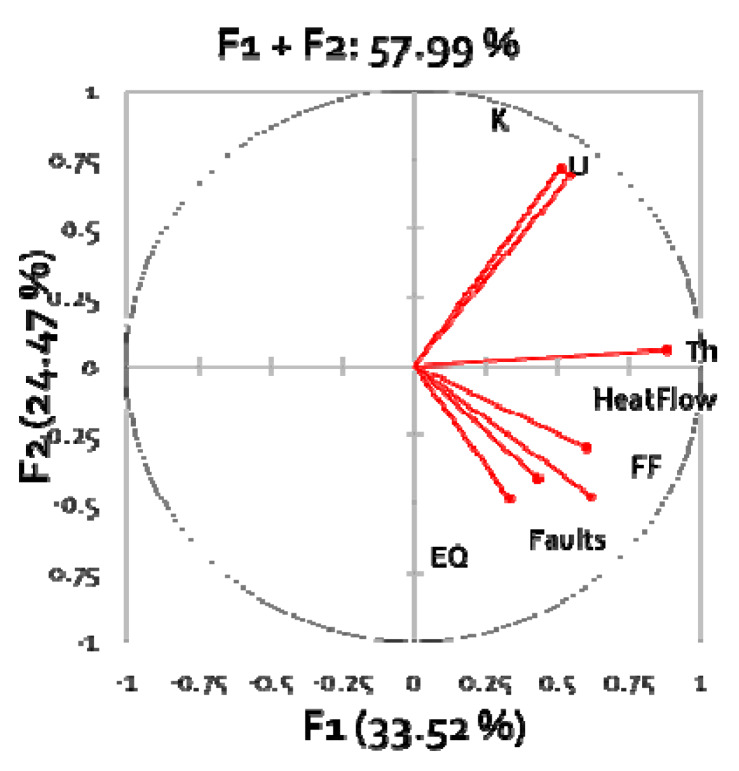
Raw PCA result. Loading plot, showing the coefficients of each variable for the first component versus the coefficients for the second component. This graph shows which variables have the largest effect on each component. Percentages: Explained variance (in percentages) of first principal components F1 and F2 (From [30]).

**Figure 10 ijerph-17-04134-f010:**
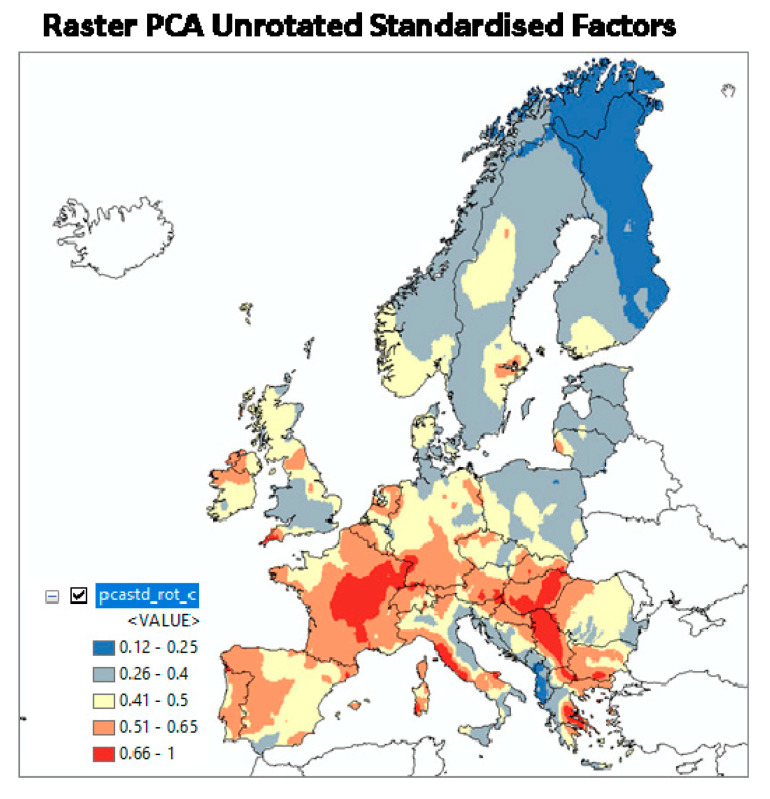
GRHI map derived from the first principal component (From [30]).

**Table 1 ijerph-17-04134-t001:** Taxonomy of GRHI definitions. See Section 3.3 for more details.

	A “Geogenic”	B “Optimal~IRC”
(1) “global”	[54] physical reasoning leading to the radon availability number (RAN). [55,56,57] classification of factors related to lithology, soil characteristics, relief, soil cover, sealing of the ground, and other. [58,59] cross-classification of control factors SRC, permeability. [60] Classification of lithology, U concentration, and presence of features like faults and mines. [61,62] Classification of geology and ADER. [31] Principal component analysis (PCA) of various geogenic factors. [63] regression of Neznal-GRP vs. soil U concentration, IRC, and ADER. [64,65] Integration of hierarchical multicriteria analysis and GIS, SMCDA, incorporating various geogenic variables.	[14] Neznal-GRP, method: regression IRC vc. SRC and permeability classes [42,66] Neznal-GRP, application [67] logistic regression of IRC vs. lithological classes, TGDR, permeability, faults. [32] ML regression IRC vs. many geogenic predictors (geochemistry, soil properties etc.) [68] Regression IRC vs. many geogenic predictors (geochemistry, soil properties etc.) Multivariate classification through contingency tables: a possible method, no references so far.
(2) “local”	[69,70] multivariate classification: U.S. EPA approach; missing values allowed. [47] transfer models to estimate GRP from various geogenic quantities. [49] weighted mean of classified quantities, see Figure 1. [50] correlation of various geogenic quantities with Neznal-GRP.	[50] correlation of various geogenic quantities with IRC

**Table 2 ijerph-17-04134-t002:** Compliance of approaches A and B and variants (1) and (2) with the desired properties of the GRHI.

	A + (1)	A + (2)	B + (1)	B + (2)
I consistent	yes	difficult	yes	difficult
II exhaustive	no	yes	no	yes
III simple	some not simple	relatively simple	some not simple	relatively simple
IV predictor IRC	to be checked	to be checked	yes	yes

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
