# Peer review of "Development of a Geogenic Radon Hazard Index—Concept, History, Experiences"

_ijerph, 2020, doi:10.3390/ijerph17114134_

Round 1

Reviewer 1 Report

General

Line 66. “The purpose of this paper is to summarize the current (early 2020) state of conceptualization and definition of the GRHI”, (geogenic Rn hazard index). The details and techniques used for this conceptualization are not given and must be, including the magnitude of the data sets used.

The calculation of GRHI is shown in Fig.1. Figure 1 is unclear, details of the techniques must be given including.

  1. The variable TGDR is not defined.
  2. The criteria for the ranking A to D .
  3. State each variables’ ranking.
  4. State each categorical and continuous variable.
  5. Explain how variable weights are determined.
  6. Is a weight given for data quality. If not, why not.
  7. How many grid cells are available for use, and n, the average number of samples in a grid for each variable.
  8. Do the number of grids cover a country or only a small fraction of their area.
  9. What hazard is the hazard index identifying. Explain a value of the hazard index such as 0.5.

Figures 8 and 9 show results of two GRHI calculations. Explain the cause of the large differences in GRHI. Give the fraction of the mapped areas shown in the Figures that have significant data for the variables.

Conclusions, lines 636-662. The conclusion seems to be that the task proposed is difficult.

Mapping geogenic radon and GRHI does not appear to map variability of hazard (approach A) but a probability value from predictions made with some selected variables.

Specific

line 19. Change “good parts” to “large parts”

line 22. Do these calculations “assist radon mitigation policy, allocate resources, to tune regulations or focus on where mitigation is needed with priority.” Explain.

line 171. This statement must be explained by example,”A variant without resorting to classification of variables has been shown”.

Line 539. AML requires a better definition

Fig 1. TGDR not defined.

Figs. 8, 9. Describe the factors/charateristics of the hazard index 0 to 1. What does hazard index 1.0 propose.

Fig 10. Explain F1 and F2

Fig. 11 Explain the values of pcastd.

Author Response

thank you for your suggestions. We tried to take care of all, providing an explanation where we did not. Please find below our responses to your comments printed blue. In the revised ms., modified and amended sentences are printed red, except trivial corrections. Also linguistic modifications are not marked.

Apart from addressing reviewer comments, a few errors were corrected and references added.

Comments and Suggestions for Authors

General

Line 66. “The purpose of this paper is to summarize the current (early 2020) state of conceptualization and definition of the GRHI”, (geogenic Rn hazard index). The details and techniques used for this conceptualization are not given and must be, including the magnitude of the data sets used.

This paper intends to give an overview of concepts. Going into very detail of each method would require a book. The same applies for detailed description of datasets.however they are all referenced.

The calculation of GRHI is shown in Fig.1. Figure 1 is unclear, details of the techniques must be given including.

    The variable TGDR is not defined.

    The criteria for the ranking A to D .

    State each variables’ ranking.

    State each categorical and continuous variable.

    Explain how variable weights are determined.

    Is a weight given for data quality. If not, why not.

    How many grid cells are available for use, and n, the average number of samples in a grid for each variable.

    Do the number of grids cover a country or only a small fraction of their area.

    What hazard is the hazard index identifying. Explain a value of the hazard index such as 0.5.

The scheme shown in the figure only serves to illustrate a general approach. We quote it because it was the first trial of several structurally similar ones, all based on calculating weighted means of predictors. It has never been used as such to generate an actual map, but later elaborated in different ways (as referenced in the section). We quote it because in our opinion, it shows nicely the structural idea. As it is intended to visualize a concept, there are no technical details to be discussed, nor specific datasets to be described. Therefore, it has been included in section 2 (concepts), but not in the results section (4). Over the years, it served as kind of guideline towards constructing the GRHI, even if different techniques were applied which emulate the basic concept of Fig.1 (some discussed in sec. 3.3).

We hope that the modifications of the text, including linguistic improvement, help clarifying our intention. In particular, the paragraph preceding the figure (now lines 179 ff.) was modified to improve clarity.

Figures 8 and 9 show results of two GRHI calculations. Explain the cause of the large differences in GRHI. Give the fraction of the mapped areas shown in the Figures that have significant data for the variables.

1) We are not sure to which maps the reviewer refers. Figures 8 and 9 in original counting (now 7 and 8) refer to maps generated by ML and MR. These are indeed very similar; quite surprisingly given very different methodology. The PCA map, now Fig.10 (formerly 11), is different. The main reason is that it shows a different concept of GRHI, namely one following type A, instead type B, as do the former. PCA is an exploratory method aimed to extract variability components of a given dataset, whereas regression a confirmatory one, i.e. aimed to explain a given response.  .

2) By construction, all areas have significant data for the variables.

Conclusions, lines 636-662. The conclusion seems to be that the task proposed is difficult.

Mapping geogenic radon and GRHI does not appear to map variability of hazard (approach A) but a probability value from predictions made with some selected variables.

It is not supposed to map variability, but to construct a quantity (the GRHI) such as to optimally honour variability of geogenic controls of Rn.

Specific

line 19. Change “good parts” to “large parts”   

changed, but differently; now “many parts”

line 22. Do these calculations “assist radon mitigation policy, allocate resources, to tune regulations or focus on where mitigation is needed with priority.” Explain.

The sentence has been modified:

“Since geogenic radon is an important predictor of indoor radon, knowing the local potential of geogenic radon can assist radon mitigation policy in allocating resources and tuning regulation to focus on where it is needed with priority.”

line 171. This statement must be explained by example, ”A variant without resorting to classification of variables has been shown”.

We rephrased the sentence. What we want to say is that in that variant, numerical quantities are not classified (A to D in the example), but used as they are.

Line 539. AML requires a better definition

The text has been modified.

Fig 1. TGDR not defined.

Added in the legend and in the acronyms list.

Figs. 8, 9. Describe the factors/characteristics of the hazard index 0 to 1. What does hazard index 1.0 propose.

0 = smallest, 1 = highest. A note added in the text.

Fig 10. Explain F1 and F2

has been explained in the text.Additionally, legend expanded.

Fig. 11 Explain the values of pcastd.

removed

Reviewer 2 Report

[General comments]

This manuscript is a review paper of the previous geogenic radon hazard index (GRHI) approaches with discussing and summarizing the concept and definition of the GRHI. And the author provided GRHI map on the European level using various statistical techniques. The authors explained statistical methods in detail but too long. The paper is well written and well organized. I suggest minor revision before to be accepted. Here below detailed comments.

[Major comments]

  1. The explanations of PCA, ML, and MR….are too long. In particular, Line 373–416 part is a very general explanation of PCA as you mentioned in the manuscript (The method has been established for long and refinements have been developed). Furthermore, Line 458–468 part is also. I suggest that long explanation parts could be reduce to 1 or 2 sentences if the method is well known.
  2. In general, 222Rn can be easily dissolved in groundwater (222Rn activity in groundwater is generally more than 100–1,000 times higher than those of seawater or surface water. I think 222Rn activity in groundwater is higher than that of soil air.). In the manuscript, you mentioned “hydrogeology”. Do you consider groundwater as geogenic 222Rn source? Additionally, 220Rn, one of radon isotopes, is especially, shows high activity in some closed-systems although it has a very short half-life (56 s).

[Specific comments]

- There are some mistakes in ‘Figure number’; For example, there is no Figure 4.

- Also, there are some typos.

- Check the parentheses and the reference form in the manuscript.

Line 20: “~ in few ~” è “~ in a few ~”

Line 384: “~ i.e.~” è “~ i.e.,~”

Line 389: “~ Figure 4 ~” should be changed.

Line 415: “Detailed description ~” è “Detailed descriptions ~”

[End]

Author Response

Dear reviewers,

thank you for your suggestions. We tried to take care of all, providing an explanation where we did not. Please find below our responses to your comments printed blue. In the revised ms., modified and amended sentences are printed red, except trivial corrections. Also linguistic modifications are not marked.

Apart from addressing reviewer comments, a few errors were corrected and references added.

Comments and Suggestions for Authors

[General comments]

This manuscript is a review paper of the previous geogenic radon hazard index (GRHI) approaches with discussing and summarizing the concept and definition of the GRHI. And the author provided GRHI map on the European level using various statistical techniques. The authors explained statistical methods in detail but too long. The paper is well written and well organized. I suggest minor revision before to be accepted. Here below detailed comments.

[Major comments]

    The explanations of PCA, ML, and MR….are too long. In particular, Line 373–416 part is a very general explanation of PCA as you mentioned in the manuscript (The method has been established for long and refinements have been developed). Furthermore, Line 458–468 part is also. I suggest that long explanation parts could be reduce to 1 or 2 sentences if the method is well known.

We shortened the text, but only the section about PCA. Motivated by comments of the other reviewers, these technical sections even had to be expanded.

    In general, 222Rn can be easily dissolved in groundwater (222Rn activity in groundwater is generally more than 100–1,000 times higher than those of seawater or surface water. I think 222Rn activity in groundwater is higher than that of soil air.). In the manuscript, you mentioned “hydrogeology”. Do you consider groundwater as geogenic 222Rn source? Additionally, 220Rn, one of radon isotopes, is especially, shows high activity in some closed-systems although it has a very short half-life (56 s).

Hydrogeological classes serve as proxies for permeability, i.e. Rn transport properties. Indeed, Rn concentrations can be high in groundwater, e.g. for a groundwater temperature of 10 °C, which is typical for central Europe, the partitioning ratio air : water is 3:1. This means Rn concentration in air is higher compared to groundwater for the same mineral material. Nonetheless, groundwater can serve as source of Rn since it can flow from a geogenic high Rn background area into a geogenic low Rn background area.

However, this effect of imported Rn via groundwater is limited in most cases due to low groundwater flow velocity: Flow velocity of groundwater is most often < 1m/d, i.e. during 5 Rn half-lives the groundwater is transported only 20m. This general picture might be different for karstic systems where groundwater flow velocity is usual higher and for areas with significant upward groundwater flow. However, there are no databases about Rn in groundwater or depth to water table, except for local studies, i.e. not covering Europe, as needed for our objective.

Concerning geogenic thoron (Tn), this is an interesting topic which has been investigated very little. On the other hand, due to its short half-life, it can be assumed that geogenic Tn contributes little to indoor concentration. Anyhow, this is a topic which deserves to be studied further, but is outside the scope of this paper.

[Specific comments]

- There are some mistakes in ‘Figure number’; For example, there is no Figure 4.

Has been fixed.

- Also, there are some typos.

The ms. has been checked for typos and linguistic errors.

- Check the parentheses and the reference form in the manuscript.

We tried to rectify, but the rules are very complicated, so that we are not entirely sure whether all is now correct.

Line 20: “~ in few ~” è “~ in a few ~”

“in few” is correct

Line 384: “~ i.e.~” è “~ i.e.,~”

“i.e.,” is correct

Line 389: “~ Figure 4 ~” should be changed.

done

Line 415: “Detailed description ~” è “Detailed descriptions ~”

done

Author Response

Dear reviewers,

thank you for your suggestions. We tried to take care of all, providing an explanation where we did not. Please find below our responses to your comments printed blue. In the revised ms., modified and amended sentences are printed red, except trivial corrections. Also linguistic modifications are not marked.

Apart from addressing reviewer comments, a few errors were corrected and references added.

Notes about IJERPH-794715

General comments

The topic addressed by the manuscript is relevant for environmental science: a review of the conceptualization and trials of the evaluation of the geogenic radon to indoor radon exposure is of great scientific interest. However, the manuscript is not acceptable in its present form, and it needs to be revised before publication. My main concerns are the following.  

  1. a) The manuscript presents a great deal of acronyms (e.g. GRHI, GRP, SRC, ML, IRC ecc.). Some of them (i.e. GRHI and GRP) are mentioned already in the introduction but without a clear explanation of their meaning. Many terms seem synonyms, but it is not clear if they are or not. The reader could be disoriented and could find difficult the comprehension of the differences among the used terms. Considering the aims of a review paper, such as providing an exhaustive summary and understanding on the topic, I suggest supplying a glossary of the acronyms and terms used, specify the meaning and the unit of measurements.

A list of acronyms has been added.

  1. b) Section 2.4 is a key section of the paper but it is not exhaustive. The different approaches and variants are not detailed in a clear way.

We hope that this has now been improved

 Table 1 is not self-explanatory and should be enriched with more descriptions. Authors should consider adding one ore more figures for helping the understanding of the topic with pragmatic examples of methods applications.

The table has been expanded and corrected.

The problem is that the paper is already very long. Adding more text and figures would probably not be accepted neither by the editor nor by most readers. Those who would really get into the subject in technical detail must be asked to consult the references.

Moreover, I suggest indicating the variants as “1” and “2” in order to prevent misunderstandings.  

done

  1. c) The paragraph about “Multivariate estimation” (Line 366-371) is not exhaustive and it suffers from lack of references and of a more complete theoretical background.

This paragraphs is about classification; 2 examples added. Actually the method has not much of theoretical background, but is largely based on experience.

 In radioactivity mapping, multivariate algorithms (such as Collocated Co-CoKriging, Kriging with variance of measurements) have an important role for studying the spatial distribution of radiological variables (e.g. total activity, uranium concentration) based on gamma-ray spectroscopy surveys. Authors should widely comment this approach.  

We do not agree in this point. We do not deal with spatial modelling, i.e interpolating point data, but with estimation of a new quantity (GRHI). Spatial modelling has an auxiliary role only, i.e. for pre-processing variables or post-processing the GRHI. This has been mentioned in the ms., but it is certainly not a main topic.

In general captions are insufficient, more details should be added to help the comprehension of the figures and tables; acronyms and variables definition (e.g. TGDR, Z, X) are absent The correct use of the English language should be improved, the wording should be revised to improve clarity (see for example specific comments).  

Specific comments

Line 118. I suggest to specify that “non geogenic factors” are anthropogenic factors.  

sentence modified

Line 128 and line 287: equations have formatting problems (subscript an superscript) and typos (“:” after “=”).  

Equ.s are correct in the submitted doc.

Line 131. Please better specify the meaning of SRC 0 , giving some references for zero and “small values”.  

done

Line 132. It is not clear what “the numerical value” refers to.  

done

Line 142. Please provide a reference for GRP data available in the mentioned countries.  

done

Line 147-151. Please rephrase the paragraph, it is not sufficiently clear.  

done

Figure 1. The number in “Soil properties” box should be 6. Please define “TGDR” in the caption.  

done

Line 215. The reference of Figure 2 is wrong.  

fixed

Line 280. Please provide a more exhaustive title of the section.  

Title modified; paragraph added at end of section.

Line 328. Please define better the meaning of “radon characteristic”  

Sentence slightly modified.

Line 334. Please rephrase the sentence, it is not sufficiently clear.  

done

Line 342. “Figure 3” should be “Figure 2”.  

Numbering of figures has been corrected

Line 358. It is not clear the meaning of “IRC (observed) – model”. It is a subtraction between the observed and modeled values? If yes, please write “IRC (observed) – IRC (modeled).  

done

Line 360- 363. Please rephrase the sentence, it is not sufficiently clear.  

The section has been largely rewritten

Line 370. “Target variable can BE”  

done

Line 374. “from possibly many potentially”: please rephrase it.  

done

Line 523. “100 covariates Y”: please explain better.  

slightly modified

Line 639. “…to use?; how…”: Please remove “;”.  

done

Round 2

Reviewer 1 Report

A Geogenic Radon Hazard Index for Europe if reasonably accurate is an important contribution to wide area potential hazard from radon and based on geogenic radon . However, the authors must always include the information that theirs is a calculated potential hazard from geogenic radon not a predictor of hazard. Line 21 must be deleted or changed, stating “Geogenic radon has been shown to be an important predictor of indoor radon.”

The first mapping of regional radon geologic potential was published by Gundersen et al. USGS in 1992 (ref below). It was based largely on bedrock and climate data. Gunderson (ref 70) later published geologic estimates along with indoor radon measurements and for a high potential area measurements ranged from average to high radon concentration. The author's calculations must  always be stated  as estimated “potential”.

Line 322 section 3.2 The databases include 7 data sets for the data base. The data sets are mentioned but it remains vague what the actual bases(es) consist of and the size of these bases. Is every European country included in the data, line 699 says databases cover most countries? Explain. Figures 7 and 8 do not appear to agree with figure 10. Explain. Some specific details about the data sets and the countries are needed. Perhaps a table of basic information.

Specific

  1. 105, 121 etc. ReV should be defined in the acronyms.
  2. 163. A meeting cannot be referenced as a country
  3. 576. Define the specific hazard in Hazard Index 0 to 1. What is the numerical risk of index 1?

Fig. 2. Approach A, Including indoor radon concentration to predict indoor radon concentration variability certainly yields predictable results. Explain.

Fig. 3. Must be clarified with a specific example. Predictors Z0 , Z0’ etc. Proxys Z1, Z2. This should explain how predictor Z0 can affect both Proxy Z1 and Z2

Reference should be included

Gundersen, L.C.S.; Schumann, R.R.; Otton, J.K.; Dubiel, R.F.;Owen, D.E.; Dickinson, K.A. Geology of radon in the United States. In: Gates, A.E.; Gundersen, L.C.. Geologic controls on radon. GSA Special Paper 271, 1-16; 1992. Boulder, CO: Geological Society of America.

Author Response

Dear reviewer,

thank you for your suggestions. We tried to take care of all, providing an explanation where we did not. Please find below our responses to your comments printed blue. In the revised ms., modified and amended sentences are in correction mode, as suggested by the editor.

Regarding linguistic issues, one of the authors is native English speaker, so that we think that we are on the safe side.

reviewer 1

A Geogenic Radon Hazard Index for Europe if reasonably accurate is an important contribution to wide area potential hazard from radon and based on geogenic radon . However, the authors must always include the information that theirs is a calculated potential hazard from geogenic radon not a predictor of hazard. Line 21 must be deleted or changed, stating “Geogenic radon has been shown to be an important predictor of indoor radon.”

This sentence does not exist in the 1st revision of the ms.

But apart from this, I think that we have a terminological misunderstanding. Risk is posed by exposure to indoor Rn, not by geogenic factors. As also explained in the ms., risk is the result of various factors, among them geogenic ones. These do not define the risk, but the potential risk from geogenic factors, i.e. the geogenic hazard.

Because the risk pattern is largely controlled by the pattern of geological factors (i.e. not the size of risk, but its geographical pattern), mapping the geogenic hazard is an important task – as also explained in the ms.

Of course the geogenic Rn hazard index can be a predictor of hazard – hence its name. In order to serve as IRC or risk predictor, a further step is necessary, namely linking GRHI or GRP with IRC or exposure; but this is a separate task, not subject of this paper. However, the GRHI shall be tailored (concept B) such as to optimally serve as such predictor (requirement IV).

The first mapping of regional radon geologic potential was published by Gundersen et al. USGS in 1992 (ref below). It was based largely on bedrock and climate data. Gunderson (ref 70) later published geologic estimates along with indoor radon measurements and for a high potential area measurements ranged from average to high radon concentration. The author's calculations must always be stated  as estimated “potential”.

Line 322 section 3.2 The databases include 7 data sets for the data base. The data sets are mentioned but it remains vague what the actual bases(es) consist of and the size of these bases. Is every European country included in the data, line 699 says databases cover most countries? Explain.

The detailed description of the databases is found in the references. Repeating this in the article would inflate it enormously without – in my opinion – any information benefit for the objective of the paper. For example, not all countries participated in the geochemical FOREGS and GEMAS projects. This is deplorable, but we have to live with it for variant (1) and can only hope that in the future these databases will eventually be completed. However, this lacunae is certainly an argument for a variant (2) approach, which would allow missing predictor values.

Figures 7 and 8 do not appear to agree with figure 10. Explain.

This has been said in the text. The reason is that it represents a different approach (A vs. B). A sentence added in the ms.

Some specific details about the data sets and the countries are needed. Perhaps a table of basic information.

See above. The paper is already very long, and indeed much more could be said. Have mercy with the editor!

Specific

    105, 121 etc. ReV should be defined in the acronyms.

done

  1. A meeting cannot be referenced as a country

true. Modified.

  1. Define the specific hazard in Hazard Index 0 to 1. What is the numerical risk of index 1?

Again: we are not talking about numerical risks. The numerical quantity is the result of some algorithm, whose size is of no importance. For convenience, this is transformed into [0,1], but any other scaling is possible, as explained in the ms.. Risk could be something like lung cancer incidence, but this not at all in the scope of this paper.

A GRHI map is one which shows the spatial dynamic of geogenic factors, condensed into one quantity, but NOT risk levels.

Fig. 2. Approach A, Including indoor radon concentration to predict indoor radon concentration variability certainly yields predictable results. Explain.

Approach A yields results which may serve to predict IRC. For B, this is so by default, because of its definition. Including IRC (or IRC normalized to standard anthropogenic conditions, such as ground floor, building with basement, etc., in order to get rid of statistical noise caused by variability of anthropogenic factors) in A will probably improve its capacity to predict (normalized) IRC. But qua construction it will not be an optimal predictor a priori. Exactly therefore, B has been conceived.

Fig. 3. Must be clarified with a specific example. Predictors Z0 , Z0’ etc. Proxys Z1, Z2. This should explain how predictor Z0 can affect both Proxy Z1 and Z2

Example: Z1 = ADR, Z2 = GRP; Z0’” and Z0” could be fallout and Th, Z0 could be U, Z0’, permeability. I think that this is self explaining, however a short piece of text has been inserted. 

Reference should be included

Gundersen, L.C.S.; Schumann, R.R.; Otton, J.K.; Dubiel, R.F.;Owen, D.E.; Dickinson, K.A. Geology of radon in the United States. In: Gates, A.E.; Gundersen, L.C.. Geologic controls on radon. GSA Special Paper 271, 1-16; 1992. Boulder, CO: Geological Society of America.

Thanks for the suggestion! This is a useful report. I amended it to existing ref. 69 instead of inserting as new ref., to avoid tedious renumbering of the subsequent part of the reference list. Indeed pioneering work on the subject comes from the US and the Czech Republic.